# A Guideline for Implementing a Robust Optimization of a Complex Multi-Stage Manufacturing Process

**Francesco Bertocci** [1,*], **Andrea Grandoni** [2], **Monica Fidanza** [1] and **Rossella Berni** [3]

1  R&D Global Transducer Technology, Esaote Spa, Via di Caciolle, 15, 50127 Firenze, Italy; monica.fidanza@esaote.com
2  Supply Chain, Procurement, Manufacturing, Esaote Spa, Via di Caciolle, 15, 50127 Firenze, Italy; andrea.grandoni@esaote.com
3  Department of Statistics Computer Science Applications "G. Parenti", University of Florence, Viale Giovanni Battista Morgagni, 59, 50134 Firenze, Italy; rossella.berni@unifi.it
*  Correspondence: francesco.bertocci@esaote.com

**Featured Application:** This research represents a key-study devoted to the continuous improvement of manufacturing processes leading to a concrete advantage for companies. The proposed guideline allows to enhance the process capability and the product quality, including the achievement of a robust process by limiting the variability and the scraps. The suggested guideline is also illustrated through a pilot study related to US probes for medical imaging, although it can be applied to any manufacturing product/process.

**Abstract:** In the industrial production scenario, the goal of engineering is focused on the continuous improvement of the process performance by maximizing the effectiveness of the manufacturing and the quality of the products. In order to address these aims, the advanced robust process optimization techniques have been designed, implemented, and applied to the manufacturing process of ultrasound (US) probes for medical imaging. The suggested guideline plays a key role for improving a complex multi-stage manufacturing process; it consists of statistical methods applied for improving the product quality, and for achieving a higher productivity, jointly with engineering techniques oriented to problem solving. Starting from the Six Sigma approach, the high definition of the production process was analyzed through a risk analysis, and thus providing a successful implementation of the PDCA (plan-do-check-act) methodology. Therefore, the multidisciplinary analysis is carried out by applying statistical models and by detecting the latent failures by means of NDT (non-destructive testing), i.e., scanning acoustic microscopy (SAM). The presented approach, driven by the statistical analysis, allows the engineers to distinguish the potential weak points of the complex manufacturing, in order to implement the corrective actions. Furthermore, in this paper we illustrate this approach by considering a pilot study, e.g., a process of US probes for medical imaging, by detailing all the guideline steps.

**Keywords:** robust optimization; plan-do-check-act (PDCA); problem solving; statistical modeling; SAM (scanning acoustic microscopy); ultrasound probe; continuous improvement; Six Sigma; multi-stage manufacturing process

## 1. Introduction

Advanced robust process optimization techniques are widely adopted in the manufacturing field, in order to enhance the productivity and to improve the quality of the processes by limiting the effect of fluctuations, noises, and scraps. At the same time, the more and more competitive and stringent industrial scenario requires: (i) keeping production costs to a minimum, and simultaneously, (ii) the maximization of the products' quality [1]. Every existing company must be able to maintain and increase the efficiency and the effectiveness with the aim to maintain their competency and competitiveness at

high level [2]. Quality control methodologies, such as Six Sigma, created at Motorola in the 1980s, have been adopted by leading companies throughout the world for quality and process improvement aiming to enhance an organization's competitiveness. Six Sigma [3–5] is a quality methodology which reduces variations in a process, and it helps to lower the cost of product as well as process. In the last three decades, the Six Sigma approach has been implemented by many industries and most of them obtained fruitful benefits [6].

Nevertheless, in the Industry 4.0 scenario, and for a complex multi-stage production process, the application of the Six Sigma methodology needs to be expounded and integrated by also considering other techniques. Moreover, the engineers, aiming towards the continuous improvement of the manufacturing, should consider the risk level mitigation for each complex production stage by identifying and understanding the weak points. The support of advanced statistical methods must be integrated [7,8]. To this end, it must be noted that, more recently, the statistical methods suggested and applied for a robust process optimization have been widely extended by considering: (i) the wide range of experimental designs [9], including optimal designs; (ii) the possibility to evaluate heteroscedasticity and random effects (not only limited to noises) by applying, in the modeling step, a class of statistical models specifically used to treat/study complex situations [10]; (iii) achieving the optimum of the response and the corresponding optimal setting (optimal solution) of the experimental variables by analytical optimization [11], or, alternatively, by comparing several optimal solutions in a decision process that can also include costs and other economic issues. In [12], the decomposition of the mean square error (MSE) is performed also according to several error components related to external elements, such as costs or consumer loss. To this end, in this paper we aim to address a guideline (also called procedure), in which the whole quality process improvement is outlined by considering engineering issues and statistical methodologies, also evaluating features and recent developments in the literature.

These days, robust design optimization (RDO) methods have obtained a great deal of attention for improving engineering performances and manufacturing processes [13]. The concept of robust design was primarily introduced by G. Taguchi [14–18]. Robust design allows us to make a product and/or a process insensitive to environmental noises and/or other internal or external sources of variability [19]. The key issue for obtaining a RDO resides in setting the control factor levels to be insensitive to noise variables, often involved in the studies as random effects [20]. There are many potential uncertainties that can arise and influence the practical application in a manufacturing scenario by including process failures, material properties variability, environmental variability (noises), and human-factor dependency [21]. In addition to uncertainty and noise effects, the choice of the right statistical model must consider other issues, which might be imposed in problems under real operational conditions such as dynamic situations, multi-response cases, discrete and continuous data, physical constraints to design factors, performance costs, computational complexity [10–12]. Therefore, the control of these factors and sources of variability becomes critical when the performance of the product must be maintained within the desired specifications. The optimization problem is the core of a robust design study, and a key concept is to opportunely plan the experimental design, in order to successfully transfer the engineering knowledge and measurements to statistical variables [22].

In [23], six factors were involved and supposedly influencing the performance of the tooling during the manufacturing process: productivity, quality, time, machine, operator, and color of the insulating material, but the research work focused on the efficiency of the tooling die-nozzle. The engineers stopped the analysis by considering a limited number of factors both for the conditioning due to the daily engagement in the manufacturing, and for the need to rapidly solve the problems; a simple statistical approach was applied involving the analysis of variance and the Tukey test [24].

In this study, we aim to address the key tools and to outline the methodology, defined at each step, for a successful robust optimization. The study also integrates the statistical methods within the well-known techniques and application in various manufacturing

industries used in engineering as the plan-do-check-act (PDCA) cycle (also known as the Deming cycle). The latter is used for improving the existing processes in an industrial scenario, in which the lean manufacturing principles and practices are widely adopted [25,26]. Defects are considered the greatest waste in the manufacturing systems, and they negatively affect the delivery times, costs, and quality of products [27]. Kaizen activities are deeply implemented for rapid problem solving, which allows for improving the quality and the productivity in manufacturing industries [28,29].

A major challenge for the continuous improvement of ultrasound probes for medical imaging is how to choose and define the responses and how to identify the factors (influencing the responses) that represent the manufacturing process trend. The identification of the sources of process variability requires the medium–long term for implementing the whole study with the aim to know in depth the potential problem or weak points of the existing manufacturing process.

An ultrasound (US) probe whose behavior has been opportunely settled according to a robust design concept will have more chances to have a good performance and to be mostly stable, in presence of fluctuations and noises during the operative life.

In the literature, the current situation shows a gap when considering studies related to the continuous improvement for multi-layered structures composed of silicone, piezoelectric material, and epoxy resins, with an overall thickness in the range 300–1000 µm. Therefore, in this paper, and also relating to the case study, the aim is twofold: (i) we attempt to address the issues for the valid application of the robustness design principles, which remain a key challenge for the manufacturing field, and (ii) the proposal is applied to manufacturing products currently in production, i.e., ultrasound probes for medical imaging, which allows us to better highlight the key steps of the general guideline, a general guideline in which the engineering issues and statistical methodologies are combined, and successfully applied, also evaluating the recent developments in literature.

In fact, the suggested procedure is related to the quality improvement activities by combining statistics and engineering, in order to continuously improve manufacturing process operations, and to reduce resource waste by enhancing the goods.

To this end, the pilot study can be a guideline for applying the advanced robust design optimization methods to US probes for medical imaging, but not limited to these products. In a highly competitive and demanding electronics and biomedical market, the research studies devoted to the continuous improvement of the manufacturing process can be implemented through advanced statistical methods, i.e., statistical modeling, jointly with the advanced instrumentation for defect detection, i.e., scanning acoustic microscopy (SAM). The latter is a robust non-destructive testing (NDT) technique that provides an efficient solution for quick identification and location of failures in the bulk of US probes.

The rest of the paper is organized as follows. Section 2 describes the US probe for medical imaging and the manufacturing process. Section 3 presents the methodology consisting of the PDCA cycle, the statistical modeling, and the SAM analysis. Section 4 shows the obtained results. Section 5 includes the discussion. Section 6 contains the conclusions and recommendations derived by the suggested methodology.

## 2. Materials

Ultrasound diagnostic technology is generally related to imaging of biological tissue by using a probe. The latter is typically placed on the body surface or internal to a body lumen of a patient in a selected imaging region. Today, the ultrasonography is becoming more and more useful for the management of COVID-19 (Corona virus desease-19) with respiratory involvement and for the rapid assessment of the severity of SARS-CoV-2 (Severe acute respiratory syndrome coronavirus (2) pneumonia [30,31]. The safety, repeatability, absence of radiation, portability, and low cost are the main characteristics for lungs inspection. The tracking of the evolution of disease, the monitoring of lung recruitment maneuvers, and the fast response for making decisions related to weaning the patient from ventilatory support are some of the benefits of ultrasound probes for medical imaging.

The US probe geometrical precision and the guarantee of high process capability by limiting the materials damage are continuous tasks that the engineers must chase. In fact, the multi-stage manufacturing process of ultrasound probes for medical imaging is composed by the assembly of layers with different physical and chemical properties and by mechanical dicing of the multi-layers structure with the thickness around some micrometers. The choice of materials involved in the US probe is very important for assuring at the same time the best image quality during the medical diagnosis, the manufacturability and stability of the process, and the cost reduction for manufacturing.

### 2.1. Ultrasound Probe for Medical Imaging

The study is related to a convex array probe for medical imaging (Figure 1) devoted to abdominal imaging and women health diagnosis, i.e., C 1–8 [32]. This convex array is composed of 192 elements with frequency in the range 1–8 MHz along a curvature of 50 mm radius. The US probe is realized according to the criteria used to design commercially available piezoelectric probes [33]. The probe (Figure 1a) includes an US transducer (Figure 1b), which transmits US pulses and receives US echoes reflected from the tissue and placed in contact with the skin of the patient through the acoustic lens. The transducer then receives ultrasonic waves reflected from the region and converts the received waves into electrical signals that are processed to form a diagnostic image. The cables and the housing allow for the direct connection to the US system and for the best plank ability of the sonographer, respectively.

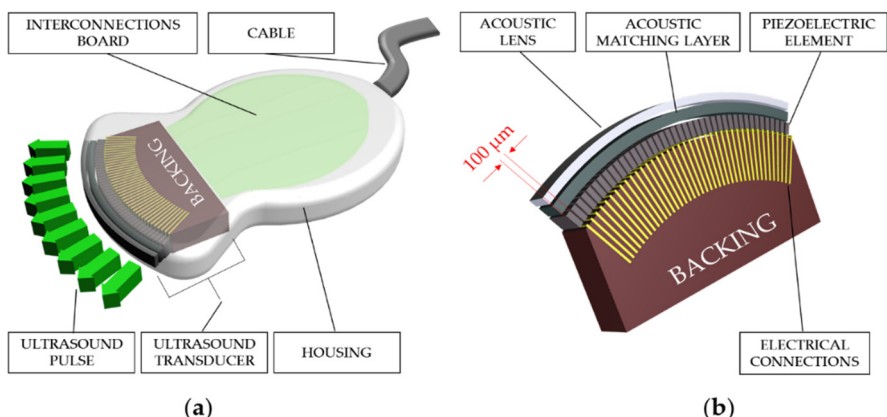

(**a**)　　　　　　　　　　　　　　　　　　　(**b**)

**Figure 1.** Typical convex array ultrasound device: (**a**) ultrasound (US) probe; (**b**) US transducer.

The backing material acts both as a support on which the piezoelectric elements are mounted and as a damping device for the back-travelling acoustic wave, to minimize reverberations and ringing. A metallic, typically aluminum, block acts as a support for the backing material. The remaining layers are the acoustic matching layers located between the piezoelectric elements and the front of the US probe. These layers are used for providing a progressive stepwise adaptation, which also allows to maintain a sufficiently large bandwidth for the passing ultrasound pulses.

The piezoelectric elements convert the electrical signal provided by the interconnections board into an acoustic wave and vice versa. Each one of the piezoelectric elements forms an emitting and receiving transducer. Piezoelectric elements are fabricated in single crystal material PMN-PT (lead magnesium niobate-lead titanate).

The latter is the main component that is divided by means of mechanical dicing, in order to realize an array of elements. The ultrasonic transducer is characterized by 192 piezoelectric elements that are arranged in line 96 on the odd side and 96 on the even side.

### 2.2. The Manufacturing Process

The growing demand for geometrical precision and the guarantee of high process capability by limiting material damage is a continuous task that engineers must chase. The manufacturing process of ultrasound probes for medical imaging is composed of 23 stages (Figure 2a) and it requires homogeneous layers with specific thicknesses (from some microns up to 500 μm) by the bonding operation between materials with different thermal expansion coefficients and densities. The bonding stages of the layers requires homogeneous bonding lines with specific thickness (up to 1 μm) [34]. The bonding process is realized in a controlled environment, 24 ± 1 °C with relative humidity less than 50%. Factors such as roughness of the components, curing time and temperature, deposition technique, and viscosity of the epoxy resin are very important for guaranteeing not only the adhesion between layers, but also a bonding line less than 1 μm and the homogeneity of the deposition.

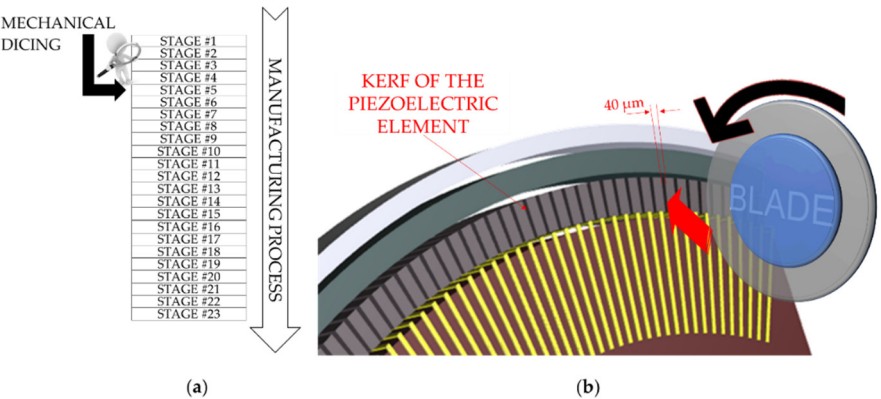

(a) (b)

**Figure 2.** The manufacturing of an US probe: (**a**) the stages of the process; (**b**) the mechanical dicing of the US array.

Other essential operations, having a high impact on the final quality of the US probe, are the micro-soldering of the electrical interconnections and the mechanical dicing operation, that create the element of the piezoelectric array. The operations are completely manufactured at the site of Esaote S.p.A. located in Florence (Italy). The production line for this type of US probe is organized in batches, in which different components are assembled through step by step processes. The manufacturing process based on a batch size of 4 pieces had been principally chosen due to the blade wear (dicing saw—DAD3350—DISCO—Tokyo, Japan) devoted to mechanical dicing for this specific product. This stage is critical for the manufacturing process, and the fundamental requirement consisting in strong and homogeneous adhesion between materials must be satisfied. The major additional variability source for the generation of micro-cracks, chipping, or delamination [35,36] is provided by mechanical dicing stress (Figure 2b), considered a very critical operation for the manufacturing of the US probe. The factors of feed rate (mm/s), spindle revolution (rpm), diamond grains size and the density of grains in the blade, and the quality of the blade cooling water avoiding the impurities [37] are important for obtaining an US transducer without cracks or delamination. The kerf (Figure 2b) allows for the electrical separation (electrical insulation) of the piezoelectric material for the generation of the piezoelectric elements array. The separation is provided by the blade of mechanical dicing: for C1–8 probe the kerf is 40 ± 2 μm.

The high number of process stages (#23), as reported in Figure 2a, needs to predict, in advance, the detection of failures in order to decrease the costs of the waste material, and the manufacturing labor costs. Moreover, it should be underlined that: (i) the analysis here carried out allows to reveal the faults in advance at stage #5 (mechanical dicing), instead of through the electrical measurements performed at stage #15 (capacitance test); and (ii) the causes of the cracks, as shown in the non-destructive technique applied at stage #3

(bonding piezoelectric element on backing), are further stressed during the mechanical dicing. However, these causes can be related to the bonding operations (performed in the early manufacturing stages), but they require a deep investigation by means of the fault detection analyses (Section 3.1) and SAM (Sections 3.3 and 4).

The quality of the US transducer is important to maintain the same limited variability in terms of acoustic output and image quality. The workers must perform specific and critical operation linked to the experience and the knowledge about the assembly of components with different type of characteristics. Process engineers are more and more focused on the implementation of automation solutions in order to reduce worker dependency.

## 3. The Methodology for Continuous Improvement: Robust Process Optimization

Manufacturing companies transform the raw materials and components that they receive from their suppliers by assembling them in order to obtain a finished product or sub-assembly, which must be delivered to the final customer just in time with respect to the delivery plan, without defects and in compliance with the technological requirements [21], including the mechanical and electrical specifications.

However, even today, defective products and components are present in the manufacturing industry, and this is a critical situation that companies in this sector are asked to face day by day. The proper care in the design, the materials selection, and product manufacturing are often not optimized for guaranteeing the manufacturing yield due to factors not considered a priori or due to the human factor [23]. In a highly competitive and demanding electronics and biomedical market, robust non-destructive methods for quality control and failure analysis of electronic components within multi-layered structures are strongly required. The robust process optimization will be useful for improving the quality of the products, and for measuring the responses that are relevant for improving the manufacturing process.

In what follows, the advanced methodologies are illustrated through the pilot study related to the US probes for medical imaging.

### 3.1. The PDCA Cycle Implementation

The PDCA cycle approach helps the company to identify the major root causes that limit the quality of the manufacturing process and to improve the know-how of the manufacturing process. Following Taguchi's philosophy [15,17,38], an inter-disciplinary team composed of engineers and physicists of R&D (research and development) of the US probes, mechanical designers together with the production and the people in charge of data collection on MES (manufacturing execution system) platform was assembled. For improving the production of an existing process, the initial step consisted of the identification of the weak points, assessing a classification of the risk level-based critical phases with the aim to obtain benefits through corrective actions in a short time.

For complex manufacturing such as US probes, it becomes obvious that the improvement of process efficiency needs a medium–long term analysis, due to critical phases of the process manufacturing, in which different materials and components are involved. Furthermore, some stages are worker-dependent.

SIPOC (suppliers, inputs, process, outputs, customers) [39,40], Pareto chart [41], COPQ (Cost of poor quality) [42], brainstorming [43], FMECA (Failure mode, effects and criticality analysis) [24,44], CTQ (Critical to quality) matrix [45], control charts [46], design of experiments [4,47], cause and effect diagram [48], and process capability [49] are well-known tools and techniques in the engineering field. The latter have been used during the PDCA cycle implementation aiming to bring an effective solution of a problem and to provide benefits for the industrial scenario. Nevertheless, when dealing with many technical issues, it is not trivial to obtain valid and efficient results, also considering the issues due to the collection of data, and the identification of causes and related effects.

Therefore, the manufacturing process (Figure 3) was analyzed not only through the process map (Figure 2a), but also using fault tree analysis (FTA) and the process failure

modes and effects analysis (PFMEA), top-down and bottom-up techniques, respectively (Figure 3a). These latter are methods applied for root cause analysis and for potential failure modes, whose combination was used for the validation and for the assessment of electric/electrical devices in the railway and aircraft systems [50] and for the risk analysis of safety-critical engineering systems [51].

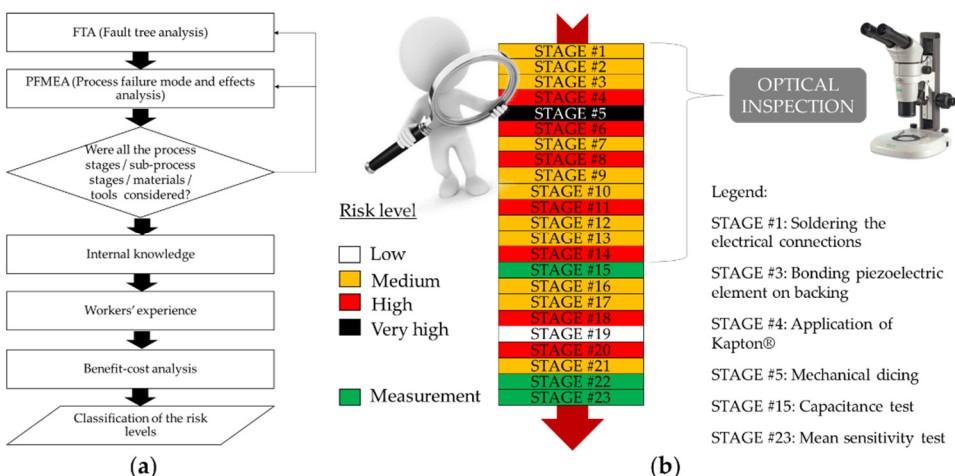

**Figure 3.** High definition of the process. (**a**) Risk level classification flowchart; (**b**) risk level assignment.

The implementation of FTA and PFMEA was developed according to [52] and [53], respectively.

In the context of the manufacturing process of ultrasound (US) probes for medical imaging, the implementation of the fault tree (logical diagram) started from the two main top events (undesired events), that are "capacitance test—FAIL" (stage #15) and "mean sensitivity test—FAIL" (stage #23) with the aim to search for the possible causes of such events.

More specifically, the capacitance test is the measurement of the capacitance of each piezoelectric element. The mean capacitance of the 192 piezoelectric elements must be in the range (350–400) pF.

The mean sensitivity for the US probe, that is the average for all array elements, named *Sensitivity$_j$*, and defined as follows:

$$Sensitivity_j = \frac{V_{RX_j}}{V_{TX_j}} \qquad (1)$$

In Formula (1), the index *j* relates to the number of piezoelectric elements, $V_{RX}$ is the peak amplitude of the received echo, and $V_{TX}$ is the peak amplitude of the transmitted ultrasound pulses. $V_{TX}$ and $V_{RX}$ are automatically measured through the measurement system for the electroacoustic (pulse echo) characterization [54].

The deductive analysis through FTA investigates the combination of failures of the process stage reaching the FAIL. In fact, the detailed analysis of the logical and temporal relationships leads to more details about the description of the possible failures of each sub-process and process stage starting from the undesired top event.

At the same time, many different workers, employees, and specialists were involved in the brainstorming meetings comprising the specialists of manufacturing stages (i.e., micro-soldering, bonding, and mechanical dicing) for implementing PFMEA. The latter provide more details about the description of the potential failures and their consequences by considering the mechanical tools used for the operations, materials, and components. The implementation of PFMEA allows for prioritizing the stages requiring improvements and optimization in short terms, too. In this case, once the corrective actions have been

applied, graphical analyses, i.e., control charts, are utilized for monitoring and measuring the effectiveness of the improvement in the process.

Applying these two complementary methods provides deeper semi-quantitative information than the separate application for the medium–long term improvement of the manufacturing process. In fact, the essence of the continuous improvement and the challenge for the process efficiency need to be medium–long term, due to critical phases of the manufacturing, in which different material are involved.

Furthermore, the analysis proposes a ranking of risk levels based not only on the combination between FTA (determination of probability of failure, P) and the PFMEA study (multiplication of occurrence (O), detection (D), and severity (S) devoted to the general evaluation of risk priority number, RPN), but also considering the internal knowledge of facing the process stage (internal knowledge, IW), the workers' background (workers' experience, WE), and the benefit-cost analysis (BCA). The approach is to transform the qualitative information into quantitative data, in order to share and to assign a value for internal knowledge, workers' experience, and BCA. In this context, the internal know-how and the workers background can evaluate the critical issues of potential failures/defects that cannot be detected through optical inspection or by means of electrical or mechanical measurements in progress in the production process.

Therefore, a score from 1 (highest) to 10 (lowest) is used for IW, WE, and BCA, which, combined with P and RPN (normalized from 1 to 10), allows for a risk level in the range (1–10,000).

The classification starts from the minimum risk, e.g., from low (1–2500) to the maximum risk very high (7501–10,000), passing through medium (2501–5000) and high (5001–7500) risks. The assignment of colors and the four levels of risk levels allows for helping not only the engineers but also the statisticians, who are completely unaware of the technical problem in any depth. The combination of these methods provides the high definition of the process based on the risk level assignment (Figure 3b).

### 3.2. The Statistical Modeling

When considering the application of statistical methods specifically defined for the robust process optimization, undoubtedly the two main key-points are the two-step procedure, and the concept of robust design, initially defined by Taguchi [17].

When dealing with statistical methods and modeling for a robust process optimization, the most widely applied methodology is the response surface methodology (RSM) [55], essentially of the I and II order. RSM was firstly suggested as an alternative method with respect to Taguchi's parameter design [16], in particular considering two seminal studies: (i) the dual response approach [56], and in 1992 the proposal of the combined array [57], where the robust design concept found the first and effective valid implementation. Moreover, starting from 1990s and more recently, all literature on this subject has been developed considering: (i) the problem of heteroscedasticity, and/or the inclusion of noises and/or random effects, among the others [58,59]; and (ii) the specific planning of experimental designs that are the most suitable for evaluating fixed and random effects for a robust design modeling and optimization, such as the split-plot design [20,60].

In this specific case study, we consider a theoretical surface:

$$\eta = f(x_1, \dots, x_i, \dots, x_k), \tag{2}$$

where $\eta$ is a surface function of *k* input or independent variables $x_i$ (i = 1, ..., k).

Starting from Formula (2), the local approximation in Taylor Series within an $I(x_0)$ allows for obtaining (and estimating) the response surface models of *d = 1st* and *d = 2nd* order, by assuming that the function *f* is continuous and derivable of order d + 1, to this end all the $x_i$ must be quantitative and continuous variables.

It must be noted that the 2nd order RSM is expounded with respect to the 1st order model by adding the first order interaction terms (interaction including two factors) and the quadratic terms for all the independent variables $x_i$ (i = 1, ..., k).

Let us now consider a dependent and quantitative variable (Y) assumed as the measure of the quality of the process; let $(x_1, \ldots, x_i, \ldots, x_k)$ be the set of factors/variables involved in the study; obviously we are assuming that all the $x_i$ have an influence on the response variable (Y).

The 2nd order RS model for the general u-th experimental observation (u = 1, . . . , n), and for K = 2, is as follows:

$$y_u = \beta_0 + \beta_1 x_1 + \beta_2 x_2 + \beta_{12} x_{12} + \beta_{11} x_1^2 + \beta_{22} x_2^2 + \varepsilon_u \; u = 1, \ldots, n \qquad (3)$$

In Formula (3), $\beta_1$ and $\beta_2$ are the two unknown coefficients related to the linear effects of the model, related to the variables $x_1$ and $x_2$; $\beta_{12}$ is the coefficient related to the 1st order interaction, while $\beta_{11}$ and $\beta_{22}$ are related to the quadratic effects. It must be noted that all the unknown 2nd order coefficients, $\beta_{12}$, $\beta_{11}$ and $\beta_{22}$, are the elements of the array **B** = (2 × 2) that plays a fundamental role for the optimization; $\varepsilon_u$ (u = 1, . . . , n) is the random error.

### The Logit Model

The logistic regression is now briefly illustrated in order to outline the fundamental elements for the case study; further details related to the logit model can be found in [61].

In this kind of statistical modeling, the main feature is the nature of the response variable; in fact, in this model the dependent variable Y is binary: Y = 0,1. The set of input (independent) variables, also called explicative variables, can be formed by qualitative and/or quantitative variables.

Therefore, through the logit model we can estimate the probability of the event "success", usually coded by "1". Thus, P(Y = 1 | **X**)= $\pi$ (x) expresses the probability of success, e.g., the probability that the event (coded by "1") occurs, given the a specific set of values **X**, related to the explicative variables; for example, if an experimental design has been carried out, the set of values **X** can correspond to a specific experimental run (trial). Usually, the event coded by "1" is the event that we are mostly interested to study. In this application, the response variable is the capacitance test (where 1 corresponds to the failure of the test, while 0 means that the US probe passes the test). Through the logit model, Formula (4), we pass from a binary domain (0;1) to the Real set ($-\infty$, $+\infty$). The estimation method applied is the maximum likelihood [62].

The goodness-of-fit of the estimated logit model can be verified through the diagnostic measures/indexes or through residuals; moreover, the hypothesis test, carried out on the estimated coefficients, is the Wald Test [62].

The general logit model, for K explicative variables, can be expressed as follows:

$$\text{Logit} \; [\pi \; (x)] = \log \left[ \frac{\pi(x)}{(1 - \pi \; (x))} \right] = \beta_0 + \beta_1 x_1 + \beta_2 x_2 + \ldots + \beta_k x_k \qquad (4)$$

### 3.3. The Scanning Acoustic Microscopy (SAM) Inspection

The techniques able to contribute to the optimization of the manufacturing process, in general, are based on the disassembly of the product, in which the reverse flow does not impact the quality improvement. In fact, the reverse engineering can help for viewing the defects [21], but in many process this is no possible due to the destruction of the product. The US probes is the case in which the quality inspection requires non-destructive testing techniques (Figure 4).

The infrared thermography is influenced by ambient factors (i.e., moisture, temperature) and by different reflection coefficients of the materials, especially when the components are embedded in or bonded with each other in micro-metric range.

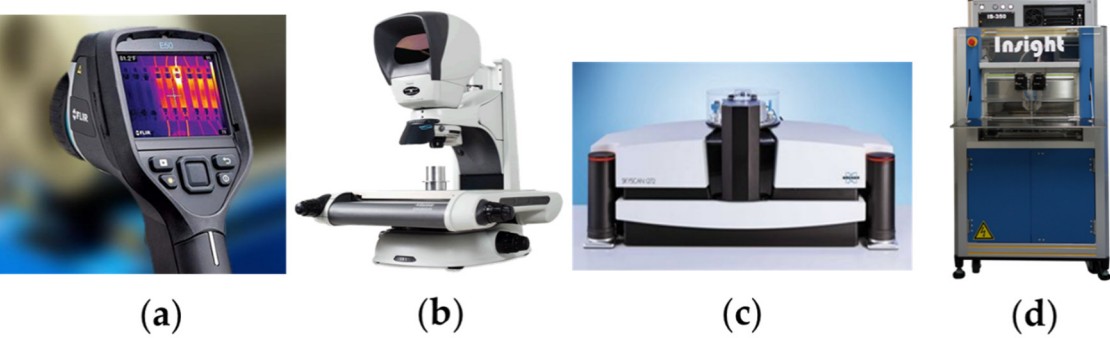

**Figure 4.** NDT (non-destructive testing) techniques. (**a**) Infrared thermography; (**b**) optical microscopy; (**c**) X-ray computed tomography (XCT); (**d**) scanning acoustic microscopy (SAM).

The optical microscopy is strongly limited to a surface investigation and X-ray computed tomography (XCT) is non-effective in case of layers of micrometers thickness, in which the resolution is a negative aspect to consider when the defects with dimension in the micrometric order must be detected. Furthermore, the composition of the PMN-PT material limits the potentiality of the X-ray-based technique due to the high content of Pb. In fact, lead acts as a shield and the result is an image without information. The authors implemented a large campaign of measurement by means of XCT analysis in order to test the effectiveness of high-resolution 3D X-ray microscopy by means of SKYSCAN 1272 (14,450 × 14,450 pixels)—BRUKER. The phase-contrast and the resolution did not distinguish the virtual slices through samples and, therefore, the defects. The benchmark led the authors to the synchrotron X-ray imaging, but the timing of scan and the data processing for an area with dimension ~900 mm$^2$ and thickness of 1 mm have been calculated more than two days per piece and the conditions for the promising in-process inspection becomes very prohibitive for the manufacturing of the products. However, the X-ray-based techniques can be effective for smaller samples under test and in case of layer-by-layer inspection.

On the other hand, the visualization of the internal structure in a non-destructive manner, and the detection of the potential effects of the failure root causes make SAM an important tool in a wide range of manufacturing process. SAM has high potential in microelectronics packaging [63] by detecting voids [64], delamination [65,66], and micro-cracks [67]. Defect inspection proved high confidence levels of failure localization and the reliability assessment in MEMS (Micro electronic mechanical systems) sensors [68].

SAM is an in-line inspection investigated in the previous study by the authors [69]. SAM analysis has been effective in non-destructive testing for the detection of failure during the manufacturing process of US probes. Therefore, this method becomes very attractive combined with less preparation time for the inspection aiming to reduce the costs of scrapped material and to determine a priori the quality degree of a product.

The challenge is to detect the potential defects on a multilayered structure with dimension of 60 mm in length and 15 mm in width. The generation of internal defects, e.g., micro-cracks and delamination, or the realization of non-homogeneous layers of epoxy adhesive in the ultrasound transducers for medical imaging are not tolerable.

## 4. Statistical Modeling and Non-Destructive Testing Results

The initial stages of the manufacturing process were certainly the most complex to analyze, also considering that the first measurement is at stage #15, e.g., the capacitance test. In fact, it is just during these initial steps that the main failure may occur, as for latent defects, risen during the construction of the US probe, even though may become evident starting from stage #5, during the mechanical dicing, which is a stage at a very high risk.

Therefore, in order to analyze these issues, a relevant strenuous collection of systematic and observational data (n = 296), based on processing cards and control charts, was carried

out. To this end, the use of MES for internal data management allowed for implementing a dataset on an electronic spreadsheet. The statistical modeling applied through SAS (SAS—Statistical Analysis System—Windows Platform) allowed for studying the quantitative response (Y) assumed as a measurement of the mean sensitivity (performed at stage #23) for the US probe.

The estimated model, based on the RSM model Formula (3), is detailed in the following expression (5); in Table 1, variable description, coefficient estimates, standard errors, and *p*-values (*t*-test) are reported.

$$\mathbf{Y} = \beta_0 + \beta_{1j} \cdot X_1 + \beta_{2l} \cdot X_2 + \beta_{3l} \cdot X_3 + \beta_{4s} \cdot X_4 + \beta_{5t} \cdot X_5 + \beta_6 \cdot X_6 + \beta_7 \cdot X_7 + \beta_8 \cdot X_8 + \beta_9 \cdot X_9 + \beta_{10} \cdot X_{10} + \beta_{11s} \cdot X_{11};$$

$$j = 1,2,3; l = 1,2,3,4; s = 1,2; t = 1,2,3,4,5;$$

(5)

**Table 1.** Model results. Response Y = Sensitivity; Formula (5).

| Parameter | Symbol | Estimate | St. Error | *p*-Value (*t* Test) |
|---|---|---|---|---|
| Intercept | $\beta_0$ | 373.19 | 40.33 | <0.0001 |
| Worker-St1-#1 | $\beta_{11}$ | 1.84 | 17.88 | 0.9194 |
| Worker-St1-#2 | $\beta_{12}$ | 6.30 | 13.08 | 0.6364 |
| Worker-St1-#3 | $\beta_{13}$ | 0.0 | | |
| Worker-St9-#1 | $\beta_{21}$ | −19.57 | 20.51 | 0.3535 |
| Worker-St9-#2 | $\beta_{22}$ | −29.87 | 39.17 | 0.4561 |
| Worker-St9-#3 | $\beta_{23}$ | −54.51 | 26.77 | 0.0576 |
| Worker-St9-#4 | $\beta_{24}$ | 0.0 | | |
| Worker-St11-#1 | $\beta_{31}$ | −41.22 | 29.42 | 0.1791 |
| Worker-St11-#2 | $\beta_{32}$ | −18.45 | 27.47 | 0.5109 |
| Worker-St11-#3 | $\beta_{33}$ | 3.49 | 24.15 | 0.8867 |
| Worker-St11-#4 | $\beta_{34}$ | 0.0 | | |
| Dicing-Saw-1 | $\beta_{41}$ | −11.53 | 6.88 | 0.1117 |
| Dicing-Saw-2 | $\beta_{42}$ | 0.0 | | |
| #Blade-usage-1 | $\beta_{51}$ | 5.38 | 10.15 | 0.6031 |
| #Blade-usage-2 | $\beta_{52}$ | −7.11 | 10.56 | 0.5098 |
| #Blade-usage-3 | $\beta_{53}$ | −9.42 | 9.59 | 0.3397 |
| #Blade-usage-4 | $\beta_{54}$ | 5.89 | 9.66 | 0.5502 |
| #Blade-usage-5 | $\beta_{55}$ | 0.0 | | |
| #days-St2–3 | $\beta_6$ | 0.97 | 4.94 | 0.8466 |
| #days-St5–6 | $\beta_7$ | 0.13 | 7.15 | 0.9858 |
| #days-St8–9 | $\beta_8$ | −0.59 | 3.14 | 0.8513 |
| #days-St10–11 | $\beta_9$ | 11.67 | 5.94 | 0.0660 |
| Trn-Flateness-St5 | $\beta_{10}$ | 4.30 | 3.28 | 0.2072 |
| Kapton®-St4 | $\beta_{11,1}$ | −28.32 | 33.45 | 0.4090 |
| Kapton®-St4 | $\beta_{11,2}$ | 0.0 | | |

The applied statistical model (5) is simpler with respect to the RSM model, Formula (3), because it does not include first order interactions and quadratic effects; nevertheless it resulted globally highly significant (model-sum-of-squares: *p* value < 0.0001) with an estimated $R^2$ equal to 0.85. Even though some of the estimated effects are not significant, some of them are relevant; by also considering the corresponding model of the analysis of variance (ANOVA) (not shown here), for each effect we obtained a relevant or an important result, measured by the F-test, for the global analysis. Undoubtedly, workers, application of Kapton®, and the number of days (among stages) play a relevant role. It must be noted that in this first step, we faced two issues: (i) we dealt with observational data; and (ii) the statistical models applied were simply regression models that allowed for estimating the main effects for the all variables involved.

Even though we did not perform an experimental design, e.g., the n = 296 observational units are US observed transducers, the study is very useful in order to perform a detailed and fruitful statistical analysis and modeling. Nevertheless, as we can observe in

Table 1, standard errors are quite large, and this implies that some estimated coefficients, even though related to main process variables, were not significant.

In addition, the statistical model (Formula (5)) includes both quantitative and categorical variables, therefore the model estimates (Table 1) are reported by considering the number of levels for each categorical variable, (e.g., the number of coefficient estimates are equal to the number of degrees of freedom of each variable). Thus, the estimation is carried out through the cornered point *criterium*, by which a categorical level is assumed as the reference level, and it is posed to "0.0" (Table 1).

The second statistical model, applied in this study, is the logit model (Formula (4)). In this specific situation, the response binary variable Y is defined considering the first response variable within the process, namely the measurement process implemented during the manufacturing process, and located at stage #15, that consists in the capacitance test. A lower or a greater mean capacitance out of the specification (350–400) pF implies the transducer is a scrap (FAIL = 1). In case of compliance, the transducer passes to the manufacturing of the next stage #16 (PASS = 0). Therefore, when considering the binary response variable Capacitance PASS/FAIL (performed at stage #15), the estimated logistic regression model for the compliance of the us probe is as the following:

$$Y_{binary} = \theta_0 + \theta_{1t}\cdot X_1 + \theta_{2j}\cdot X_2 + \theta_{3l}\cdot X_3 + \theta_{4j}\cdot X_4 + \theta_5\cdot X_5 + \theta_6\cdot X_6 + \theta_7\cdot X_7;$$

$$j = 1,2,3; l = 1,2,3,4; t = 1,2,3,4,5;$$

(6)

The likelihood ratio test obtained for the estimated logit model (6) is equal to 21.33 (14df), *p*-value 0.0933. This result allows for rejecting the null hypothesis that can be expressed through: "all the estimated effects are null or negligible".

It must be noted that in the logit model, we included several discrete variables (Table 2): amount of blade-use at five levels (estimated with regard to #1); dicing-saw at three levels (estimated with regard to #2); workers employed at Stage 9 (four workers; estimates are performed with regard to #1); workers employed at Stage 11 (three workers; estimates are performed with regard to #1). The main relevant results show that workers and the time spent among stages are significant, or almost highly significant, while the rest of the variables play a relevant role (blade, dicing-saw, and resin).

**Table 2.** Logit model results. Response Y = binary response variable; Formulas (4) and (6).

| Variable Description | Symbol | Estimate | St. Error | *p*-Value (Wald's Test) |
|:---:|:---:|:---:|:---:|:---:|
| Intercept | $\theta_0$ | −11.50 | 4.63 | 0.0131 |
| #blade-usage-1 | $\theta_{11}$ | 0.0 | | |
| #blade-usage-2 | $\theta_{12}$ | −0.39 | 0.86 | 0.6525 |
| #blade-usage-3 | $\theta_{13}$ | 0.92 | 0.89 | 0.3061 |
| #blade-usage-4 | $\theta_{14}$ | 1.90 | 1.12 | 0.0907 |
| #blade-usage-5 | $\theta_{15}$ | −2.81 | 2.08 | 0.1782 |
| Dicing-Saw-1 | $\theta_{21}$ | 0.0 | | |
| Dicing-Saw-3 | $\theta_{22}$ | 3.01 | 2.56 | 0.2401 |
| Dicing-Saw-4 | $\theta_{23}$ | 2.79 | 2.59 | 0.2820 |
| Worker-St9-#1 | $\theta_{31}$ | 0.0 | | |
| Worker-St9-#2 | $\theta_{32}$ | 3.46 | 1.86 | 0.0631 |
| Worker-St9-#3 | $\theta_{33}$ | −2.66 | 1.76 | 0.1302 |
| Worker-St9-#4 | $\theta_{34}$ | 4.10 | 1.86 | 0.0282 |
| Worker-St11-#1 | $\theta_{41}$ | 0.0 | | |
| Worker-St11-#2 | $\theta_{42}$ | 4.89 | 2.29 | 0.0325 |
| Worker-St11-#3 | $\theta_{43}$ | 4.81 | 2.28 | 0.0349 |
| #days-St5–6 | $\theta_5$ | −0.36 | 0.21 | 0.0792 |
| #days-St10–11 | $\theta_6$ | 3.56 | 1.42 | 0.0125 |
| Epoxy resin-St3 | $\theta_7$ | 1.23 | 0.97 | 0.1976 |

The statistical analysis provided enough suggestions, that implied the need to implement a design of experiments by considering the first five stages of the process, in which the

mechanical dicing is considered the highest critical level of risk. The statisticians supported the engineers in the manuscript [70] and the statistical study led to implementing SAM analysis campaign able to detect potential latent failure by inspecting the US probe during the construction (Figure 5) at stage #3 (bonding piezoelectric element on backing). In this context, a method for the detection and the visualization of latent defects on US probes for medical imaging was carried out by means of SAM (Figure 6). Therefore, we decided to scan 12 workpieces by means of X-scan analysis consisting in multiple C-scan (Figure 6a) generating a stack of equidistant 2D-images (Figure 6b) by simulating the potential introduction of this NDT method in-line of the manufacturing process. Measurements were performed by means of a commercial acoustic microscope IS-350 manufactured by Insight kk, Tokyo, Japan [71]. Samples were aligned parallel to the (x-y) working plane. The exam direction, obtained through the application of SAM, was conducted by a focused ultrasound transducer that was aligned vertically to this plane along the z-axis, and it could be moved along this axis for focusing the sound energy either onto the sample surface or inside the sample volume. The workpiece and the SAM transducer were plunged into the coupling liquid, i.e., deionized water, whose temperature was regulated and stabilized to $24.0 \pm 0.5$ °C. Ultrasound pulses with frequency of 30 MHz were generated and their reflection received by a 500 MHz bandwidth dual pulser-receiver system (DPR500, JSR Ultrasonics, Imaginant Inc., Pittsford, NY, USA).

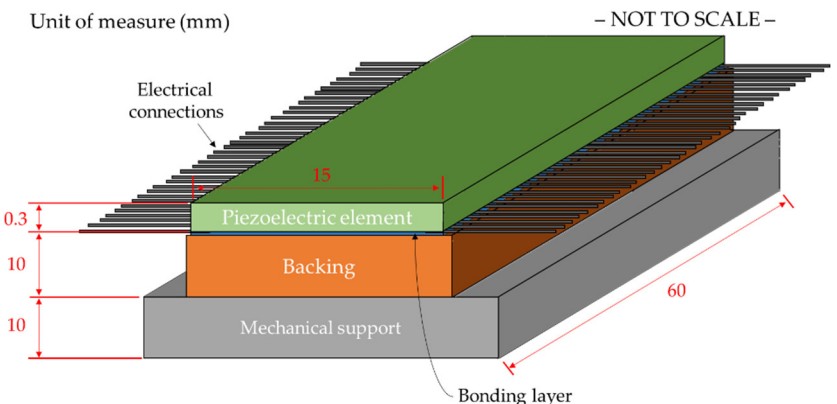

**Figure 5.** Structure of the US probe under test at stage #3.

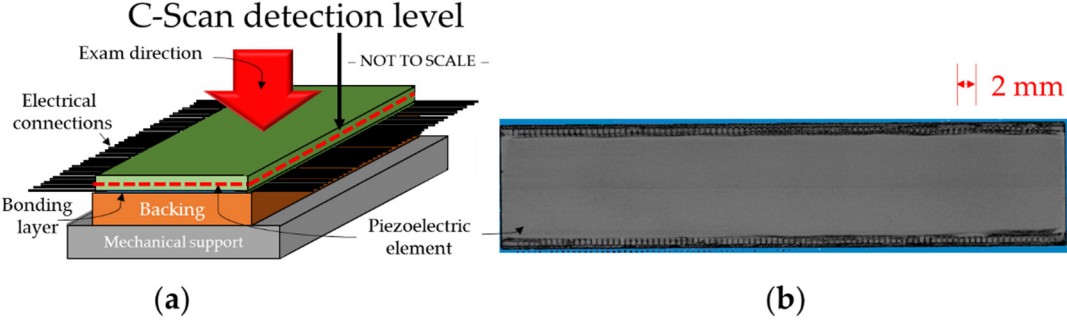

**Figure 6.** The C-scan by means of SAM. (**a**) The exam direction versus the ultrasound probe; (**b**) the image of the slice in the bulk of the piezoelectric element shows no defects.

The SAM campaign on the 12 workpieces revealed that four of them were affected by internal cracks in the bulk of the piezoelectric element (blue arrows in Figure 7). Furthermore, for the workpiece #9, a crack, located close to the central edges, was revealed (yellow arrow, Figure 7).

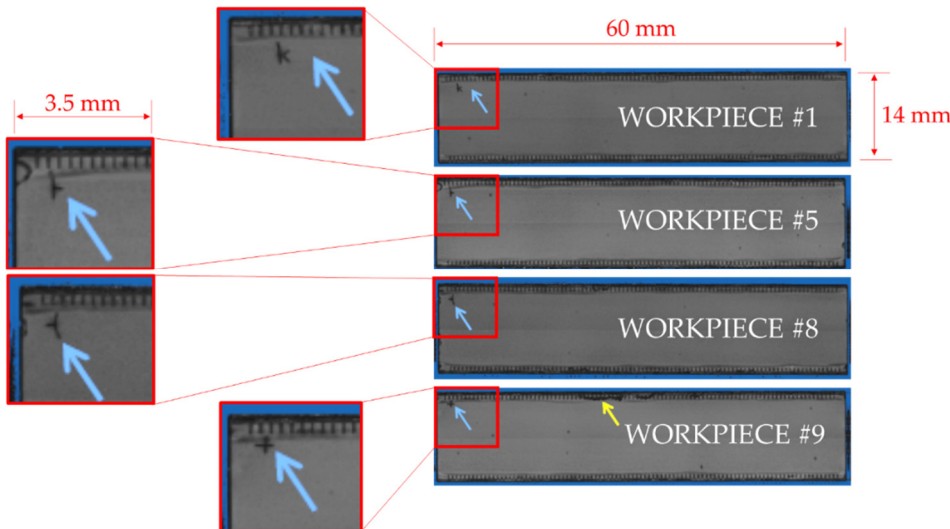

**Figure 7.** Latent defect detected in the workpieces at stage #3: crack in the US probe bulk (red rectangular). The detection is provided by SAM.

## 5. Discussion

The collaboration between statisticians and engineers led to implementing the general PDCA cycle (Figure 8). The identification of main failures and the weak points of the manufacturing process, the implementation of regression models and the detection of latent failure at the early stage of the process allowed for optimizing the overall production cycle.

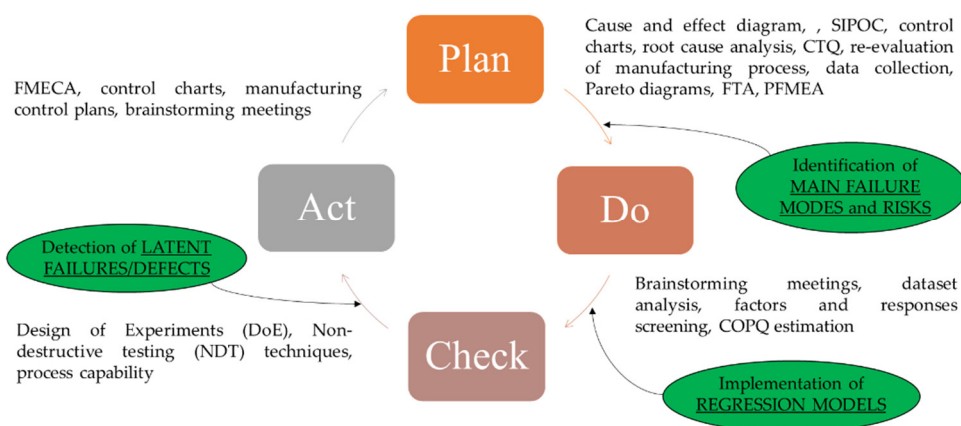

**Figure 8.** Plan-do-check-act (PDCA) cycle for the continuous improvement in US probes manufacturing process.

The successful efforts of collaboration among the engineers and statisticians became fundamental in providing clear and specific advantages articulated through the following steps:

1. The re-examination of each ultrasound probe manufacturing phase under strong critical point of view;
2. Need to add electrical and mechanical in-process measurements (several factors, several responses);
3. The identification and the analysis of factors, never evaluated by engineering, that can influence the variability of the production process;
4. The management and the analysis by considering 36 difference factors and 38 response variables (both qualitative and quantitative); and

5. The distinction between systematic, noise, and block effects for defining and planning the design of experiment [70].

Through this rigorous approach, the engineers, in agreement with the statisticians, were able to manage and to organize the analysis by considering more than 36 different factors, and 38 response variables, both quantitative and qualitative.

The two different statistical models led to the identification, by successive refinements, of the final set of factors considered most significant within the process variability. As a contribution to reduce the failure occurrences, the two selected (and applied) statistical models were used for implementing the actions necessary to improve of the production process.

The statistical results revealed:

- Significant effects of the process factors;
- The operators involved in the manufacturing play a key role in the variability of the production process;
- The need to implement a DoE (design of experiments) in the early stages of the production process for a better screening of the principal critical factors.

The continuous discussion between the engineering and the statisticians together the gradual development of know-how during this study suggested to analyze the early stages of the manufacturing process. Therefore, the NDT technique as SAM allowed for mitigating the risk level of the US probes manufacturing by intercepting latent defects during the process (Figure 9).

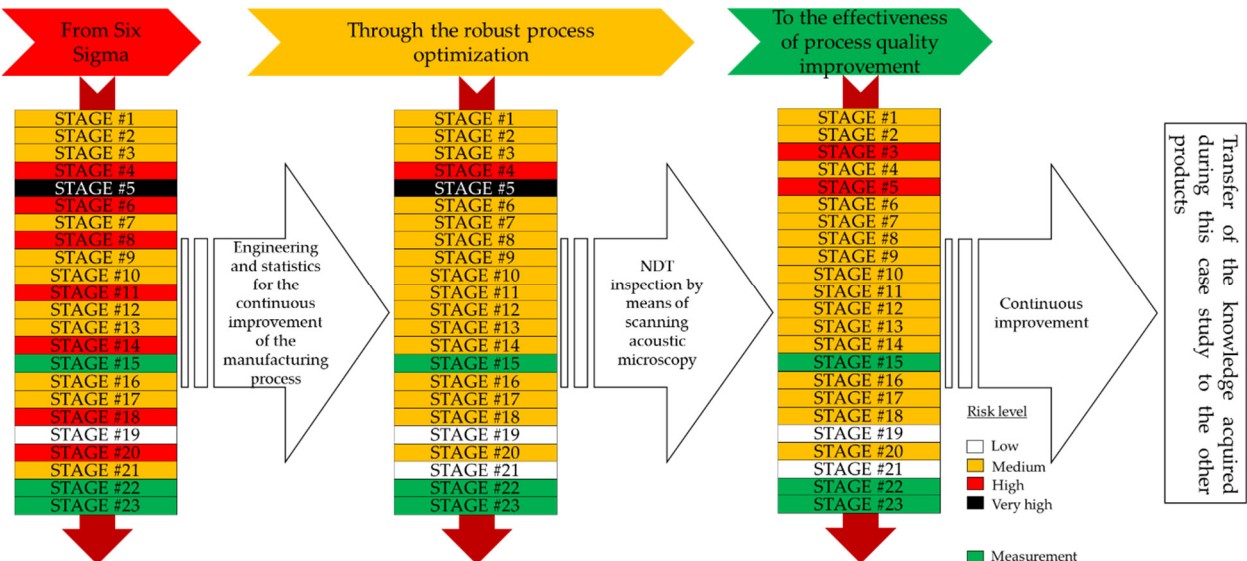

**Figure 9.** The risk level mitigation during the robust process optimization.

The robust process optimization led to a lower risk level for some stages of the process, i.e., the mechanical dicing (stage #5), but highlighting that stage #3 is also affected by latent failure detected by the SAM analysis and not recognized at priori. The damage consisting in a crack opened a tangible framework for the engineers that need to solve the problem. In a multi-stage complex manufacturing process such as US probes, the risk level is medium and this allows engineers to follow the continuous improvement approach for guaranteeing the products quality. The corrective actions applied to the multi-stage production process thanks to the continuous PDCA application have been verified and validated according to the regulations for medical products [72].

The knowledge acquired during this project allowed for transferring the methodology and the effectiveness of the process quality improvement to other products. Furthermore, the continuous improvement strategy helped the workers to take ownership for their work by reinforcing the team working and their motivations.

## 6. Conclusions

The core of the study is to illustrate, through an interdisciplinary case study involving statistics and engineering, the main steps for performing a robust process optimization and achieving the optimal setting for a response variable; moreover, our attempt is to provide, through this pilot study, a general guideline useful for all the manufacturing processes that need to be implemented following analytical procedures in order to achieve high levels of quality and process robustness. Undoubtedly, a key step is the collection of data that can be fruitful for the whole analysis. An efficient, in the sense of useful and usable, and complete dataset is the main basic element for the subsequent analyses, also when an experimental design is going to be implemented. In fact, an experimental planning necessarily requires a constructive collaboration among statisticians and engineers, in order to capture and to explain all the source of variabilities influencing the product/process. If the experimental design is adequately programmed, then the modeling and optimization step can be easily carried out, obtaining satisfactory results. Moreover, the risk classification differentiated by the colors alert all the stakeholders, at any stages and roles, also considering those who are not directly involved in the production process, aiming to increase the knowledge and the growing concentration of the weak points of the product manufacturing.

Furthermore, as stated through this pilot study, the collection of observational data can be a valid "starting point", useful for a first preliminary achieved milestone, through which the subsequent study (experimental design-modeling-optimization) can be planned successfully. Moreover, the application of two different statistical models (regression model and logit model) allows for illustrating how to choose the right statistical model when different kinds of data are available.

More specifically, the study presented the application of advanced methods for robust process optimization to identify not only the root causes of US probes manufacturing process, but also to improve the quality of the product. In this context, the multidisciplinary research was decisive for the improvement of the production process.

A new team of five dedicated operators was formed, in order to reduce the variability of the production process and increase the quality of the product. It must be noted that the involvement of human resources, e.g., workers, is the fundamental base for achieving the highest quality level of the manufacturing process.

The in-line inspection based on the SAM analysis and the proposed statistical study can open a tangible contribution for the industry 4.0, in which the real time information is the key aspect for a continuous improvement. The proposed method will benefit designers and manufacturers for achieving the optimal product performances, a high production reliability, and minimal production costs. The proposed guideline can be applied for the manufacturing of electrical and mechanical items, in which multi-stage processes are involved. Furthermore, the application of the robust optimization methods to a real manufacturing can demonstrate that the estimated statistical models, applied in this study, can effectively address practical structural problems, improving design reliability and offering broad prospects as a new design method.

Despite the numerous benefits discussed above, our pilot study is not without limitations. Firstly, we considered observational data in developing our approach. We are fully aware that it is necessary to plan an *ad hoc* experimental design in order to better perform and discover strengths and weaknesses of the investigated process. Nevertheless, it can be highlighted that all the analyses and modeling results reported in this study, can be successfully exploited for planning the following experimental design in a robust optimization context.

**Author Contributions:** F.B. conceived the guideline; R.B. implemented the statistical modeling; F.B. and R.B. wrote the paper; F.B., A.G., and M.F. performed the procedure. All authors have read and agreed to the published version of the manuscript.

**Funding:** This research received no external funding.

**Acknowledgments:** The authors would like to thank Michael Hertl and the team of Predictive Image S.A.S. (193, Chassolieres, 38340, Voreppe, France) for their precious work and support in the scanning acoustic microscopy investigations.

**Conflicts of Interest:** The authors declare no conflict of interest.

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
