# Peer review of "A Guideline for Implementing a Robust Optimization of a Complex Multi-Stage Manufacturing Process"

_applsci, doi:10.3390/app11041418_

Round 1
Reviewer 1 Report
Major comments:
The main concern I have about your results is the following:
You claim that your study allows to reduce the failure occurrence
and optimise the production cycle,
but there is clearly no attempt to do that in your work.
The defects that you show in figure 8, as you said, are given by the mechanical dicing
However, there is no attempt in your work to analyse the setting of the dicing machine
to understand why those cracks occur.
If this further analysis is done, this article may be worth publishing in a deeply revised form.
Section 3.1: it is not clear what this section is explaining
It seems like a second introduction of a list of techniques
This section should be shortened and something may be added to the introduction instead
Section 3.2 again contains a review of model.
Some of these models may be referenced in the introduction but not here.
Just report here what you actually use in your study
Section 3.3: you focus this section to describe experiments that have failed.
You should actually describe in more details the ones that were successfull (SAM)
You can remove image 4, as this is not showing anything.
Claiming that you cannot address your problem using X-Ray and Synchrotron is not correct.
The fact that you had experimental difficulties does not mean the techniques cannot be applied.
Simply the sample must be made smaller as you cannot analyse the whole device with only one experiment.
You can for instance analyse a single dicing/single layer as they are all done by the same machine
In section 4 and figure 5: you need to describe all the stages that are reported in Figure 5
and you need to specify, maybe with colors, the ones that are used as variables
for the model in table 1
Also figure 5 does not look very professional
Figure 10 is just repeating what is said in the text, it is useless
Minor comments:
In the abstract:
"allows the engineering to distinguish"
-> "allows the engineers to distinguish"
same at line 50:
"the engineering aiming", better engineers
line 85: "in deep" -> "in depth"
line 87: specify somewhere before that US is ultrasound,
otherwise reader get confused
line 104: "but not limiting to these products" -> "not limited to"
line 108: specify what NDT is, the first time you mention it
line 111: "the manufacturing process the same" ->
the manufacturing process of the same
line 162: is composed by 23 stages -> composed of
line 173: The production line for this type of Us probe -> US upper case
Figure 2 caption: The US manufacturing of US probe -> I think the first US is not needed
line 212: The PDCA cycle approach helps3333 the company -> something went wrong here
Author Response
Dear reviewer,
please check for the attached report.
Best regards.

Reviewer 2 Report
Thank you for the opportunity of reviewing your interesting article. It addresses a topic which is within the journal s scope and uses relevant literature.
Although, the paper has reached a very good level, before publication, there are several aspects that need to be addressed, namely:
At the end of Introduction section you mention that you present a pilot study that can be a guideline for applying the advanced robust design optimization methods to US probes for medical imaging, but not limiting to these products, but you did not mention it in Abstract, Methodology and Conclusion. Please, correct.
There are some minor grammar mistakes, i.e. The PDCA cycle approach helps3333 the company... Delete 3333
Author Response

(The authors gave the same response as above.)

Reviewer 3 Report
Main aim of the paper is optimization and quality improvement of manufacturing process of US probes for medical imaging. Authors used robust optimization method and PDCA cycle methodology.
Comments:
Introduction
The authors described the Six Sigma PDCA cycle method. This method is widely used not only in the Six Sigma methodology, but also in lean production (manufacturing). I recommend referring to add Kaizen, lean production too. For example:
- Smith, G., Poteat-Godwin, A., Harrison, L. M., & Randolph, G. D. (2012). Applying Lean Principles and Kaizen Rapid Improvement Events in Public Health Practice. Journal of Public Health Management and Practice, 18(1), 52-54. doi: 10.1097/PHH.0b013e31823f57c0
- Pech, M., Vaněček, D. Methods of Lean Production to Improve Quality in Manufacturing. Quality Innovation Prosperity-Kvalita Inovacia Prosperita, 2018, 22(2), 1-15. doi: 10.12776/QIP.V22I2.1096
Methodology
- Row 212: cycle approach helps3333
Discussion
- PDCA cycle is based on the Plan-Do-Check-Act stages. The last phase “Act” in this context means standardization of improved process too. Is standardization used in your case study, or is planned?
Conclusions
- Please add Limitations of your study based on used method (especially that risk in stage 3 after improvement)

Author Response

(The authors gave the same response as above.)

Reviewer 4 Report
The present work describes a research based on none destructive method for control the quality of ultrasound probes and two different statistical models, obtained in order to reveal some possible failures/defects that can occur in the different stages of their manufacturing process. The main advantage of the work is the proposed approach for combining of some engineering methods with statistical and management approaches and techniques in order to achieve improvements of the manufacturing process stages.
The work is structured in general according to the requirements of the scientific researches, but however, I have some concerns about the given statistical models and risk assessment results of the production process stages. They are as follows:
1. Only the final results of the FTA&FMEA and the p-values of the elements' significance of the derived statistical models are presented.
2. The used FTA diagram and the conducted FMEA analysis are presented too briefly, only by simple arrow diagrams on Figure 5,a.
3. The DOE plans and the chosen limits of variation of the included factors are also not shown.
4. The derived statistical models (equations 5 and 6) could be showed also in their real physical form (with numerical coefficients), but not only as theoretical stochastically models.
5. In the equation (4) parameter Vtxj on line 406 is described as "and VTX is the peak amplitude of the AW". The "AW"-abbreviation is not explained anywhere.
I truly believe that if the authors take into account above mentioned remarks, the scientific soundness of the present work will be increased significantly.
Author Response
Dear reviewer,
please check for the reply in the attachment.
Best regards.

Round 2
Reviewer 1 Report
The reviewed version of the manuscript does not address my concerns. If there is no clear investigation of the relationship between blade, dicing-saw and the fracture observed in the device, this study is not complete. The fact that cracking is caused by the mechanical process is quite obvious. Just by doing some statistics, you will not be able to understand the root cause of the problem.Author Response
Dear reviewer,
please check for the reply in the attachment.
Best regards.

Reviewer 4 Report
OK. I accept the explanations included in the authors report, and the revisions made in the paper content according to my concerns.